# A Review of the Self-Powered Wiegand Sensor and Its Applications

**Chiao-Chi Lin** [1,*] 🆔, **Yuan-Chieh Tseng** [2] **and Tsung-Shune Chin** [3]

1  Department of Materials Science and Engineering, Feng Chia University, Taichung 40724, Taiwan
2  Department of Materials Science and Engineering, National Yang Ming Chiao Tung University, Hsinchu 30010, Taiwan
3  High Entropy Materials Center, National Tsing Hua University, Hsinchu 30013, Taiwan
*  Correspondence: chiaochi.lin@mail.fcu.edu.tw

**Abstract:** Self-powered magnetic sensors are fundamental for the development of Industry 4.0, the Internet of things (IoT), wireless sensor networks, unmanned vehicles, smart cities, and sustainability. This review aimed to elucidate the working principles, materials, manufacture, output properties, and perspectives of Wiegand sensors. A Wiegand sensor is composed of a magnetic sensing wire, which is called a Wiegand wire, and a pick-up coil for the output of an electrical signal and energy. The Wiegand sensor requires an external magnetic field of about 70 Gauss to induce Wiegand wire flux changes, which, in turn, generate an output pulse in the pick-up coil. Output energy of more than 3000 nJ per single pulse (open circuit) can be harvested. The output pulse is derived from the large Barkhausen effect. Therefore, the behavior of the sensor output is independent of the triggering and sensing frequencies. The objective of this review article was to comprehensively highlight research endeavors devoted to Wiegand sensors. Furthermore, application scenarios of current research results are highlighted to find potential gaps in the literature and future contributions. Perspectives and research opportunities of Wiegand sensors are proposed.

**Keywords:** Wiegand sensor; Wiegand wire; Barkhausen effect; output pulse; self-powered magnetic sensors

## 1. Introduction

In sensing technologies, the major energy forms of measurands include thermal, mechanical, electrical, electromagnetic (EM) radiation, magnetic, and chemical energies. Sensing technologies that use magnetism possess the advantages of being non-contact and having a low input power requirement, superior energy efficiency, high sensitivity, and robust reliability against adverse environments. These advantages of magnetic sensing benefit the advancement of emerging technologies for intelligent life in terms of manufacturing, transportation, communication, the economy, and sustainability. Typical magnetic sensors include a Hall sensor, magnetoresistance (MR) sensor, flux gate sensor, magnetoimpedance (MI) sensor, superconducting quantum interference devices (SQUIDs), and the Wiegand sensor [1,2]. Depending on the sensing principles and design architecture, magnetic sensors have various pin configurations. Among all sorts of magnetic sensors (except for inductive coils/variable reluctance sensors), the Wiegand sensor is the only one that is able to be self-powered and stand alone for sensing and energy harvesting simultaneously [2].

The Wiegand sensor is named after its inventor, John R. Wiegand, for the discovery and complete investigation of its working principles, materials, and manufacture for applications in the 1970s [3–5]. A Wiegand sensor is composed of a ferromagnetic wire called a Wiegand wire that possesses special behaviors of switchable polarity and a pick-up coil that is wound around or close to the Wiegand wire for generating electrical signals while harvesting energy. An optimal external magnetic field ranging between 30 and 110 G (say 70 G) is required to induce an output pulse with a 10~30 μs full width at half maximum

(FWHM) in the pick-up coil. The external magnetic field triggers the magnetic reversal of the Wiegand wire, which is in the format of a large Barkhausen jump, generating an output pulse in the pick-up coil due to Faraday's law of electromagnetic induction. Therefore, the behavior of the sensor output depends solely on the magnetism and magnetic materials of the Wiegand wire rather than the triggering and sensing frequency of the external field. This merit of Wiegand sensors is unique.

The discovery of the Wiegand sensor did not prompt widespread research in the 1980s [6]. However, in recent years, the adoption and extensive applications of Wiegand sensors have been rapidly booming owing to the urgent need for energy-self-sufficient and maintenance-free devices for the development of sensing networks and sustainability [7,8]. The current research endeavors investigating the Wiegand sensor focus separately on materials, design, assembly configuration, circuit optimization, and application scenarios. A Wiegand sensor provides some unique advantages and properties, and hence a thorough review of the Wiegand technology as a whole enhances the broad spectrum understanding of its interdisciplinary nature. This will promote the potential of Wiegand sensors.

This review paper is divided into subsections including the working principles and the reversal behaviors of the Wiegand effect; the improvement of output properties regarding the materials, manufacture, and circuits; and the scenarios in which sensing magnetic fields occur. Then, the energy harvesting systems integrated with Wiegand sensors are listed and reviewed in terms of their practical implementation in the real world. The future trend of Wiegand sensors using various permanent magnets (PMs) to trigger an output pulse is also discussed and reported. Finally, the main niches and suggested research directions for expanding the Wiegand technology are outlined and concluded.

## 2. Working Principles and Reversal Behaviors of the Wiegand Effect

### 2.1. Working Principles

A Wiegand sensor is composed of a Wiegand wire, a pick-up coil, and an external source of the magnetic field that triggers the magnetic reversal of the Wiegand wire (Figure 1). Typical Wiegand wires have a diameter of 0.25 mm and have a core/shell configuration due to the patented method of manufacture [5]. The core and shell layers of the wire possess different magnetic hardnesses, that is, a higher magnetic coercivity ($HIST_{MAX}$) and a lower magnetic coercivity ($HIST_{MIN}$), respectively. In principle, an alternative having reversed soft/hard properties for the core/shell layers works as well.

Figure 1 elucidates the fundamental working principles for a typical Wiegand sensor [9]. The upper-left scheme in Figure 1a is a presumed initial state of a Wiegand wire, which illustrates the bi-stable behaviors of its core and shell. Such a state is usually called the "parallel state" or "reset state". The major external field ($B_{EXT}$) is longitudinal and passes through the axial direction of the wire to switch its magnetization polarity of the core and/or shell layer. The output pulses generated in the pick-up coil can be produced through the steps below:

1.  In the initial state of a Wiegand wire, both the soft and hard layers are saturated with the same polarity of magnetization (upper-left of Figure 1a). The Wiegand wire is then immersed in a longitudinal magnetic field that is capable of a cyclic polarity change (Figure 1b).
2.  When a reverse direction of the external field $B_{EXT}$ is applied and has reached a critical value, the soft layer of the wire encounters an instantaneous reversal in magnetic polarity, which induces an electromotive potential in the pick-up coil and forms an output pulse (① in Figure 1). The output pulse is independent of the rate of field change and is prominent in the pick-up coil because the generating mechanism is a large Barkhausen effect, i.e., instantaneous and gigantic magnetic polarity reversal at low external fields. At this point, the Wiegand wire is in a so-called "anti-parallel state" or "set state". This presents the 180° bi-stable magnetization of the core/shell in the wire. Usually, the output pulse is triangle-like in the time domain (Figure 1b).

3.  When the intensity of the external field continues to increase, the magnetic polarity of the hard layer also reverses, generating a minor output pulse (② in Figure 1). The Wiegand wire returns to the "reset state".
4.  When the direction of the external field is opposite and reaches a critical value, the soft layer of the wire reverses its polarity again, generating another output pulse in the pick-up coil (③ in Figure 1). The Wiegand wire is again in a "set state".
5.  When the intensity of the external field continues to increase, the magnetic polarity of the hard layer reverses again, generating a minor output pulse (④ in Figure 1). At this point, a triggering cycle has been completed, and the Wiegand wire returns to the initial state (upper-left of Figure 1).

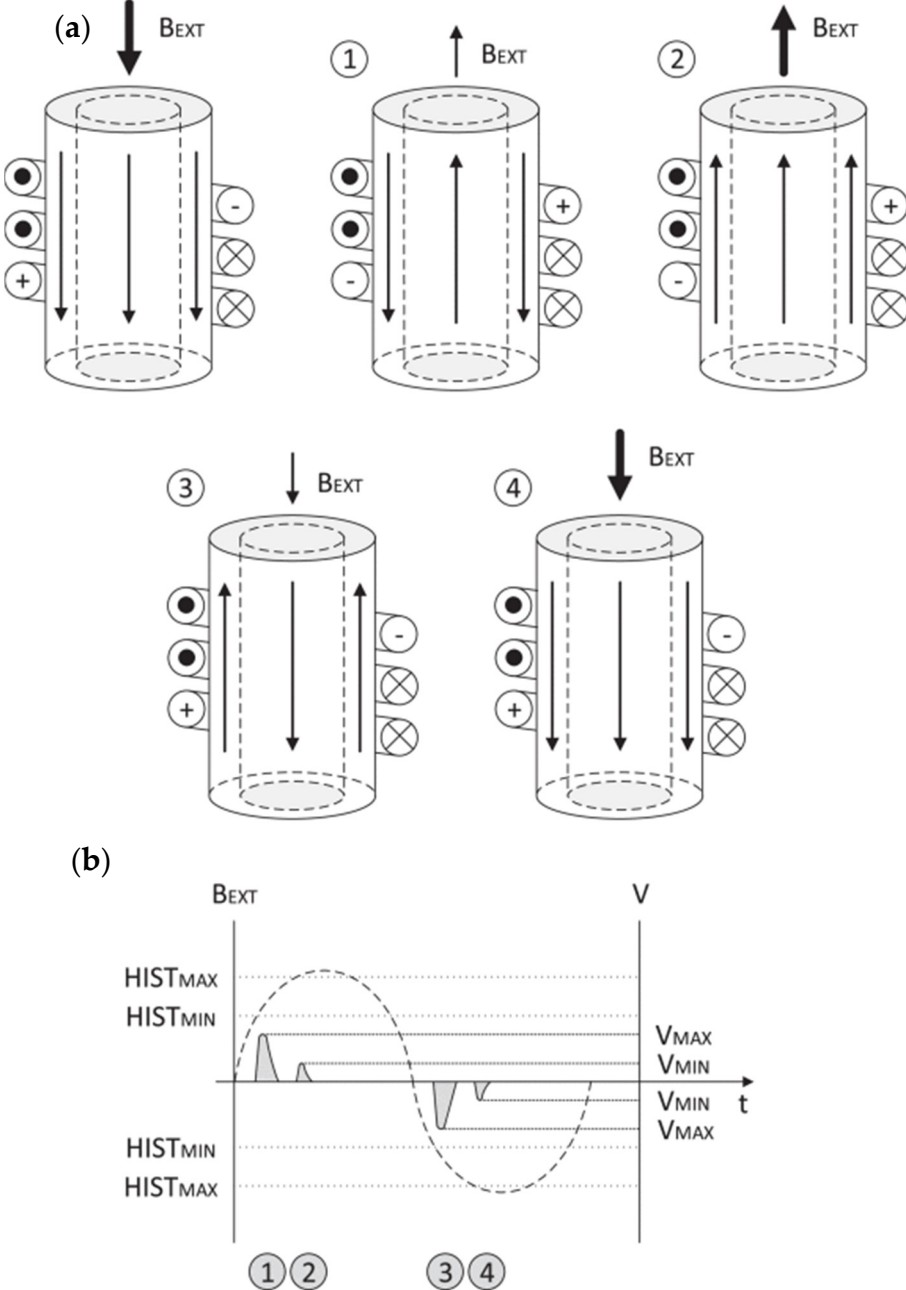

**Figure 1.** Working principles of a Wiegand sensor showing the (**a**) various states of Wiegand wires along with the corresponding external field ($B_{EXT}$) and (**b**) output pulsed voltages (V, triangle-shaped peaks) detected in the pick-up coil as a function of the cyclic changing external field (intensity indicated through dashed curve). © 2020 IEEE. Reprinted with permission from [9].

## 2.2. Magnetic Reversal Behaviors

The unique fast and enormous magnetization reversal in the soft layer of a Wiegand wire is attributed to the large Barkhausen effect [10]. The large Barkhausen effect is prominent in bi-stable ferromagnetic materials with delicate and distinct shape anisotropy, such as microwires [11–15]. The magnetic properties of microwires are strongly dependent on the materials, microstructures, and mechanical stresses through magnetoelastic anisotropy. However, the Wiegand effect is not the only one presenting the large Barkhausen behavior. The Matteucci effect is very similar to the Wiegand effect because they both present a large Barkhausen jump at the low external field in the hysteresis loops. Table 1 illustrates the comparison between the Wiegand effect and the Matteucci effect. The major differences between these two effects are the mechanisms of the domain wall (DW) formation/annihilation and motion. Nevertheless, the Wiegand effect is relatively straightforward compared with the Matteucci effect in terms of the DW kinetics.

**Table 1.** Comparison of the Wiegand effect and Matteucci effect of micro-wires, both of which produce a large Barkhausen effect [11–15].

| Properties | Wiegand Effect | Matteucci Effect |
|---|---|---|
| Typical materials | CoFeV (Vicalloy) | Amorphous Fe-based or Co-Fe-based alloys |
| Coercivity (Oe) | Tens | Few or below unity |
| Magnetization components | Axial ($M_z$) | Axial ($M_z$) and circular ($M_\varphi$) |
| DW velocity (m/s) | 500 | 600~2000 |
| Dependency of DW kinetics on the external field intensity | No [1] | Yes |
| DW propagation region for large Barkhausen jump | Bulk | Surface |
| Methods of output | Pick-up coil | Either pick-up coil or ends of the wire as direct terminals [2] |
| FWHM of the pulse by pick-up coil (μs) | 10~30 | <10 |

[1] In the working configurations of normal pulse output. [2] Depending on design and applications, e.g., GMI sensors or Sixtus and Tonks-like method [1,10,13,14].

The large Barkhausen effect in microwires originates from the antiferromagnetic-like magnetostatic coupling between soft/hard layers and/or geometrically arranged interactions [15–17]. Such a magnetostatic interaction is related to a biasing effect rather than any kind of magnetic exchange [18]. As a result, the advantageous characteristics of the Wiegand effect concerning realistic applications include the following:

1. The temperature-dependent behavior is insignificant if the materials of Wiegand wires possess a high Curie temperature;
2. The behaviors of the output pulse and sensitivity are constant despite the changing rate of the external field.

However, as there is only one magnetization component required for working with the wire shape (Table 1), erratic and irregular switching of magnetic reversal may occur if

1. Improper intensity of the external field is adopted [19];
2. The external induction line of pole pieces or magnets is not aligned well with the wire axis [19–21].

Direct observation of the magnetization states of a Wiegand wire through a magneto-optical camera indicates that a change in the external excitation field of less than 0.028 G can trigger the magnetization reversal according to [22]. This implies that a strong external field with a steep gradient of magnetic induction line disturbs the reversal behaviors of the soft layer. Hence, there is always an optimal intensity of the external field, typically 30~110 G for the maximum energy output per pulse. However, magneto-optical methods enable solely the observations of states [22,23], not dynamic behaviors. Moreover, investigations through conventional pick-up coils and minor loops (unsaturated) of hysteresis curves provide information on magnetic interactions for the whole wire, not the individual layer.

On the other hand, the first-order reversal curves (FORCs) facilitate the identification of the magnetization of various layers in the Wiegand wires. A preliminary study of the Wiegand wire length effect was reported [19,24]. More direct and time-variant characterizations when investigating the magnetic interactions and individual behaviors of soft and hard layers still await extended solutions.

## 3. Materials, Manufacture, and Output Improvement

### 3.1. Materials and Manufacture Aspects

The fundamental aspects of the technology of Wiegand sensors are its materials and manufacturing. The typical materials of Wiegand wires are NiFe alloy and Vicalloy, which is composed of a nominal composition of $Co_{0.5}Fe_{0.4}V_{0.1}$ [5,25]. Vicalloy is an intrinsically soft magnetic material with a saturated magnetization comparable to that of silicon steel (about 2 T) and a high Curie temperature of 980 °C [26]. Manufacturing processes, such as heat treatment and mechanical cold working, transfer the soft properties to hard ones. A Wiegand wire is fabricated by taking advantage of such material behaviors for producing a hard outside layer (strained by cold-working) and a soft core (unstrained). The core/shell wire is in a bi-stable status with a specific geometry [5,7,25]. However, so far only limited materials can be candidates for creating the Wiegand effect because the magnetostatic bias field strongly depends on the combination of coercivity and geometry [18].

An alternative geometrical shape for the Wiegand effect is film/substrate multilayers (Figure 2) [27,28]. Various film deposition techniques, such as sputtering, evaporation, blade coating, and electrodeposition, are very potential for manufacturing and realizing the multilayered Wiegand sensor. The variety of film deposition techniques also widens the availability of materials and the possible designs for further advancement. However, seldom the literature reports the related research result, probably due to the difficulty of process integration, configuration optimization, complex assembly, and the weak output signal to be addressed. Figure 2 outlines the major points of the wire-shaped and multilayered Wiegand sensors [27].

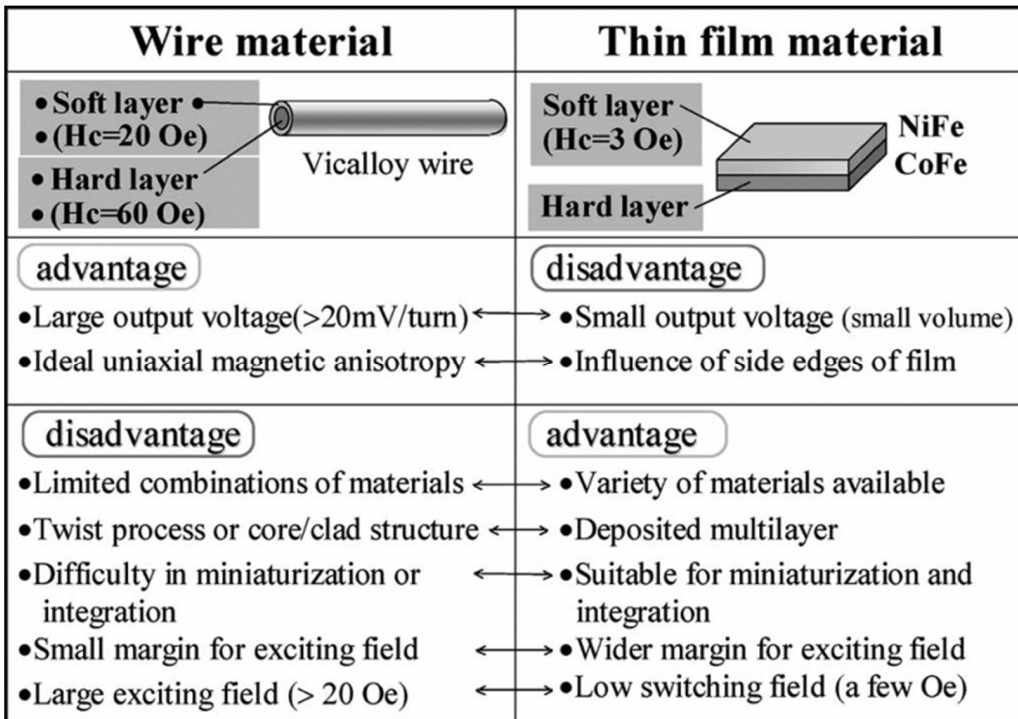

**Figure 2.** Comparison of core/shell wire and film/substrate multilayered Wiegand sensors. © 2006 IEEE. Reprinted with permission from [27].

*3.2. Strategies for Output Improvement*

To improve the output behaviors of Wiegand sensors, we must consider every aspect of the whole system. This includes the magnetism of the Wiegand wire, assembly, configuration of the ensemble, matching circuit, and energy management [29]. Current endeavors have contributed to improving the output energy and stability of the Wiegand sensors through the following:

1. Optimized Wiegand wire parameters via a systematic study of manufacturing processes, including heat treatment conditions and the stretching and twisting/torsion introduced during cold working [25];
2. Redistributed demagnetizing field of the Wiegand wire by etching the two ends of the wire [27,30];
3. Directed and guided magnetization components to align with the wire axis by adding two ferrite beads at the two ends of the wire [21];
4. A designed asymmetric external field for an enhanced large Barkhausen jump during the switching of set–reset states [2,31];
5. Arrangement and alignment concerns regarding the field source and the Wiegand wire, or concerns regarding the pick-up coil wound around the Wiegand wire [32–34];
6. Adoption of designed flux guides made of soft magnetic materials to drag and direct the magnetic field from the field sources, which are to be detected by the Wiegand wire, into the specifically located Wiegand wire through the simulation and analysis of the magnetic circuit [35];
7. Using bundled compound wires to enhance the output energy [36–39];
8. Optimized electrical circuit via equivalent circuit analysis to match the resistance/load, capacitor, rectifier, and inductance of the pick-up coil [40,41], as well as designed and optimized circuit architectures of energy management and boost converter for a one-shot operation and battery charging operation [42,43].

Some of the improvement strategies are task-oriented and dependent on the end applications. These efforts and their combinations or extensions may contribute to further development and applications of Wiegand sensors. It is worth noting that some improvement strategies may intrinsically encounter trade-offs regarding the output behaviors of a Wiegand sensor [37]. Together with the advancement of novel characterization techniques, such as FORCs [19,24], in-depth research spanning from the Wiegand wire to the Wiegand device/system can provide new opportunities in the development of Industry 4.0, IoT, wireless sensor networks, unmanned vehicles, smart cities, and sustainability.

## 4. Application Scenarios for Wiegand Sensors

*4.1. Sensing of Mechanical Motions*

Although the requirement of a triggering field with suitable intensity for switching the set–reset states of the Wiegand wire limits the capability of sensing magnetic fields, the self-powered nature and triangle-shaped pulses of the output make a Wiegand sensor ideal for sensing mechanical motions self-sufficiently [7,44,45]. In particular, the rate-independent property promotes the adoption of a Wiegand sensor in applications requiring zero-speed transducers [20,27]. Currently, the successful adoption of Wiegand sensors in industrial practices includes their use as tachometers, flowmeters, and proximity switchers. Advanced motion sensing and positioning can be realized through magnetic circuit design for event-triggering to detect, record, and count using comparator, counter, and evaluation circuits [35,46–49]. However, the arrangement, assembly, and machining precision of the components are critical to the resolution, accuracy, and tolerances of the sensing and positioning systems [20,48].

In addition, with the Wiegand sensor's unique advantages, such as robustness against adverse environments and its self-powered nature, applications in positioning drilling assembly and locating intra-body objects of medical devices were reported [50,51]. Lien et al. illustrated that positioning repeatability of 0.3 μm can be achieved by using a

Wiegand sensor [20]. Moreover, energy harvesting can be readily integrated through the Wiegand sensor based on the alternating polarities of a magnetic positioning system [35].

*4.2. Energy Harvesting*

Wiegand energy harvesters require repeatedly relative movement for switching the set–reset states. This makes the Wiegand sensor particularly beneficial for the application scenarios of vibrating devices and vehicles [7,9,32]. A stroke amplitude of 0.6 mm in a reciprocally vibrating NdFeB magnet is sufficient to trigger the output pulse for a Wiegand sensor, with the optimal stroke amplitude being ~3 mm [32]. However, designs that take both mechanical mechanisms and magnetic circuits into consideration are the keys to dynamic stability and energy efficiency [52]. The aforementioned improvement strategies in Section 3.2 shall be evaluated and selectively adopted for task-specific Wiegand energy harvesters.

The self-sufficient Wiegand sensor can assist with powering batteryless devices [29,32,35,38,42,53]. This is especially important for the applications in the fields of Industry 4.0, IoT, wireless sensor nodes, unmanned vehicles, smart cities, and sustainability. Table 2 outlines the current endeavors of batteryless and battery-assisted devices powered by Wiegand sensors. Clearly, interdisciplinary aspects, including materials, mechanical assembly, and electrical/magnetic circuits, are inevitably encountered in the research and development for these applications. Proper design and optimization of the harvesting systems with suitably scaled-up Wiegand sensors have illustrated the promising feasibilities and potentials of Wiegand energy harvesters [53].

Furthermore, Wiegand energy harvesters can realize wireless power transmission for low-power devices. This is particularly important in the scenario of implantable micro-devices [40,54,55]. Regarding health and safety concerns, the efficiency and amount of transferred energy would not be the priority. Human skin and tissues tolerate low-frequency magnetic fields better than higher ones. High-frequency magnetic fields cause tissue heating due to energy dissipation. Wiegand energy harvester's merits of frequency-independent output and miniature size trump compared with a technique such as near-field resonant inductive coupling (NRIC) [55]. This demonstrates the new paradigm of energy harvesting brought about by the Wiegand sensor.

In the end, it is worth noting that the control and management of stray fields must be considered as well. Currently, almost all the Wiegand energy harvesters utilize rare-earth permanent magnets (REPMs, mostly referring to NdFeB pole pieces) as the external field source for Wiegand sensors (Table 2). Since a Wiegand sensor relies on relative movement for switching the set–reset states of the wire, such an intrinsic nature of Wiegand devices/systems inevitably brings the problem of electromagnetic interference (EMI). Considering a triggering field intensity of around 70 G, it is not always necessary to adopt a REPM as long as the issue of mechanical interference is fixed or minor. In fact, a REPM is susceptible to corrosion issues due to the highly reactive rare-earth components. In fact, the use of REPMs does not meet the sustainability issue and will bring about stray-field and EMI issues. For Wiegand technology, engineered field lines are prior concerns instead of the intensity of the pole pieces or magnets. Hence, further exploring candidate PM materials are critical to widen the adoption of Wiegand sensors. To obtain a suitable energy product of around 3 MGOe, suggested PM materials include, but are not limited to, hard ferrite in the formats of rubber magnets or sintered pole pieces, CuNiFe, Alnico, and Co-based hard magnetic alloys produced using electrodeposition [56]. According to [35], the alternating polarity of a magnetic scale composed of ferrite is feasible as an alternative material for the external field source. Magnetization techniques and the shape effect of the pole pieces, along with the design and analysis of the magnetic circuit, are to be optimized and addressed thereof.

**Table 2.** Application scenarios for batteryless devices powered through the Wiegand devices.

| Device with Batteryless Operation | Wiegand Wire | Wire Diameter (mm) | Turns of Winding | Output Energy (nJ/Pulse) | Circuit Architecture | Ref. |
|---|---|---|---|---|---|---|
| Hall sensor sensing the magnetic field of 50~300 mT | Custom CoFeV | 0.25 | 3000 | 600 | Equivalent LR circuit with a full-wave rectifier | [38] |
| One-shot fashion and battery-charging | Commercial CoFeV | Two types, unknown | Two types, unknown | 24 for one-shot, 30 for battery charging | For one-shot: capacitor- or low-dropout-regulator-based harvesting circuit; For battery: full-bridge converter with optional battery disconnect circuit | [42] |
| Intended for FRAM | Unknown | Unknown | Unknown | 140 | Rectifier with energy buffer for data logging | [29] |
| Various IoT sensing devices | Custom CoFeV with beads at two ends | 7.5 | 8000 | >10,000 * | Ultra-wideband transceiver and harvesting circuit | [53] |
| Vibration-type generator | Custom CoFeV | 0.25 | 1000 | 7000 | Open circuit | [32] |
| Magnetic scale integrated device | Commercial CoFeV | 0.25 | Unknown | 3 | Equivalent LR circuit with a full-wave rectifier | [35] |

* Joule heating was found to be pronounced when triggering frequency > 1 Hz.

## 5. Conclusions and Perspectives

This article presents a thorough review of the working principles, materials, manufacture, output properties, and applications of Wiegand sensors. The output mechanism of a Wiegand sensor is based on the magnetostatic-bias-caused large Barkausen effect. The self-powered trait and the miniature size make a Wiegand sensor an unconventional candidate for tackling the energy-efficiency issue and sustainability concerns in developing emerging technologies. The remarkable repeatability and changing-rate-independent properties of the output pulse are the critical niches for further expanding adoptions in novel applications. Micro-scaled energy harvesting devices are the core of wireless sensing and intelligent technologies, and Wiegand sensors could play a key role in promoting the development of these technologies.

Based on the findings of this review, it is clear that there is still much room for research in applications by addressing the following:

- The output voltage depends on many relative parameters, such as the number of turns of the pick-up coil; size of the Wiegand wires; thicknesses of the soft/hard layers; and alignment between the wire, coil, field source, etc. Standardized metrics for characterizing output behaviors are necessary to compare and verify the existing and novel Wiegand sensors. However, these activities are interdisciplinary in nature. This

point is especially important when the widespread adoption of the Wiegand sensor is coming closer.

- In-depth studies on the underlying magnetism and magnetic material behaviors of the Wiegand effect have both academic and industrial contributions. A more fundamental understanding of the micro-magnetics and the micro-wire may open a new era for research activities that design, control, manipulate, and utilize the magnetostatic biasing between the soft and hard layers. Particularly, tackling the geometrical limitation of the wire's shape may widen the manufacture of novel Wiegand sensors. For example, their integration in micro-electromechanical systems (MEMS) can be realized based on the understanding of physics for the soft/hard layers interaction.
- Currently, there are only very limited materials that are able to realize the Wiegand effect. The exploration of new material processing techniques to manufacture optimally designed novel Wiegand wires or multilayered films may provide a fundamental breakthrough to overcome the technical gap of limited application scenarios. However, these activities require interdisciplinary expertise, especially regarding design optimization.
- Coupling the Wiegand energy harvesters with a supercapacitor, which forms a miniature system as a whole as an ultra-long-life power source, provides an interesting topic to study. The supercapacitor will be a part of the electrical circuit and can be easily charged by a Wiegand energy harvester.
- The design of a magnetic circuit to meet the size-critical geometry can enhance the extensive adoption of Wiegand sensors. This is also very important to tackle the EM shielding and crosstalk issues, which are commonly encountered when REPMs are used or/and as the amount of deployed heterogeneous sensors/devices is tremendous.
- To better improve the beneficial traits of Wiegand sensors, such as their miniature size and low expenditure, it is required to seek materials other than REPMs as the field source since they are susceptible to corrosion. Suggested alternatives include, but are not limited to, hard ferrite, CuNiFe, and Alnico with an energy production of around 3 MGOe.

**Author Contributions:** Conceptualization, C.-C.L., Y.-C.T. and T.-S.C.; investigation, C.-C.L. and Y.-C.T.; resources, Y.-C.T.; writing—original draft preparation, C.-C.L. and Y.-C.T.; writing—review and editing, T.-S.C.; visualization, C.-C.L.; supervision, T.-S.C.; funding acquisition, C.-C.L., Y.-C.T. and T.-S.C. All authors have read and agreed to the published version of the manuscript.

**Funding:** This work was supported by the National Science and Technology Council (NSTC), Taiwan, under the grant numbers NSTC 111-2634-F-A49-008, 111-2622-8-A49-018-SB, 110-2221-E-035-047, and 111-2221-E-035-051, and in part by MOST 111-2634-F-007-008 through the "High Entropy Materials Center" of National Tsing Hua University, Taiwan.

**Institutional Review Board Statement:** Not applicable.

**Informed Consent Statement:** Not applicable.

**Data Availability Statement:** Not applicable.

**Acknowledgments:** The authors would like to extend their appreciation to C.-K. Sung and J.-Y. Chang at National Tsing Hua University for their kind discussions.

**Conflicts of Interest:** The authors declare no conflict of interest. The funders had no role in the design of the study; in the collection, analyses, or interpretation of data; in the writing of the manuscript; or in the decision to publish the results.

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
