# Peer review of "A Review of the Self-Powered Wiegand Sensor and Its Applications"

_magnetochemistry, doi:10.3390/magnetochemistry8100128_

Round 1

Reviewer 1 Report

The article is written in comprehensible and mostly quite good English with a few minor errors or unusual constructs.

I suggest to the authors to modify the following parts of the text:

Line 35:  Typical magnetic sensors include a Hall sensor, flux gate sensor, superconducting quantum interference devices (SQUID), magnetoresistance (MR) sensor, magnetoimpedance (MI) sensor, and Wiegand sensor [1,2].

- In the order of industrial application share, i would guess the list to be more like: Hall sensor,  magnetoresistance (MR) sensor, flux gate sensor, magnetoimpedance (MI) sensor, superconducting quantum interference devices (SQUID) , and Wiegand sensor [1,2]. In other words, SQUIDs are quite limited to niche laboratory applications and far from widespread industry use.

Line 38: Among all sorts of magnetic sensors, the Wiegand sensor is the only one being able to be self-powered and standalone for sensing and energy harvesting simultaneously.

- except inductive coils/variable reluctance sensors (like the ones in automotive wheel speed sensing).

Line 47: FWHM - probably " full width at half maximum" - please define at first use

Line 54: The disclosure of the Wiegand sensor - probably "discovery" is more appropriate in this context

Line 60: Wiegand sensor owns its uniqueness, and hence a thorough review ...

- not grammatically correct. Probably should be something like: Wiegand sensor provides some unique advantages and properties, and hence a thorough review ...

Line 150: ... camera indicates the intensity differential (dBEXT) of the external field for triggering the magnetic reversal is 0.028 G [22].

- I had to study the original source (ref [22]) to understand the information.

I suggest rephrasing this as:

... camera indicates the change of external excitation field of less than 0.028G can trigger the magnetization reversal according to [22].

(that formulation is almost verbatim from conclusion of [22])

Line 322: Design of magnetic circuit to meet the case-sensitive geometry.  - I do not understand the " case-sensitive geometry ". Perhaps you mean "size-critical geometry"?

Reviewer 2 Report

The current work focuses on the Self-powered Wiegand Sensor and Its Applications. The author’s great effort into the manuscript, but minor issues should be addressed. I believe that the material presented in the manuscript is interesting for specialists in the field of magnetic sensors. It can be published in the journal after additions and revisions.

 - I felt a lack of critical assessments by the authors. The authors did not mention the research gap between the previously reported articles and the present situation Self-powered Wiegand Sensor.

In each subsection, authors should incorporate their views to mould the research in a new direction.

-It is also necessary to improve the quality of figures, in particular Figs. 1b, 2. Formulae and fonts in the figures are fuzzy and stretched; these should be rectified.

- In the introduction, the first appearance of the abbreviation should have a full definition. e.g. FWHM

- Avoid repeated, Line 11-13in the abstract is the same Line 32-35 in the introduction

-In Conclusions and Perspectives, please avoid general information lines 316-317 e,g, “Exploration of new material processing techniques to manufacture optimally designed novel Wiegand wires or multilayered films”

Please rephrase this section with the important outputs with some more detail in each point.
